# Safe and valid? A systematic review of the psychometric properties of culturally adapted depression scales for use among Indigenous populations

## Overview Review

psychometrics; Indigenous; validity; reliability; clinical utility

**Corresponding author:**
Outi Linnaranta;
Email: outi.linnaranta@thl.fi

Michelle Yang[1], Quinta Seon[2,3], Liliana Gomez Cardona[2,3], Maharshee Karia[2], Gajanan Velupillai[3], Valérie Noel[2,4] and Outi Linnaranta[2,3,5] 

[1]École interdisciplinaire des sciences de la santé/Interdisciplinary School of Health Sciences, Université d'Ottawa/University of Ottawa, Ottawa, ON, Canada; [2]Department of Psychiatry, Douglas Mental Health University Institute, Montreal, QC, Canada; [3]Department of Psychiatry, McGill University, Montreal, QC, Canada; [4]ACCESS Open Minds, Centre de recherche Douglas/Perry 3, Montreal, QC, Canada and [5]Department of Equality, Finnish Institute for Health and Welfare, Helsinki, Finland

## Abstract

**Background:** Implementing culturally sensitive psychometric measures of depression may be an effective strategy to improve acceptance, response rate, and reliability of psychological assessment among Indigenous populations. However, the psychometric properties of depression scales after cultural adaptation remain unclear.

**Methods:** We screened the Ovid Medline, PubMed, Embase, Global Health, PsycInfo, and CINAHL databases through three levels of search terms: Depression, Psychometrics, and Indigenous, following the PRISMA guidelines. We assessed metrics for reliability (including Cronbach's alpha), validity (including fit indices), and clinical utility (including predictive value).

**Results:** Across 31 studies included the review, 13 different depression scales were adapted through language or content modification. Sample populations included Indigenous from the Americas, Asia, Africa, and Oceania. Most cultural adaptations had strong psychometric properties; however, few and inconsistent properties were reported. Where available, alphas, inter-rater and test–retest reliability, construct validity, and incremental validity often indicated increased cultural sensitivity of adapted scales. There were mixed results for clinical utility, criterion validity, cross-cultural validity, sensitivity, specificity, area under the receiver operating characteristic curve, predictive value, and likelihood ratio.

**Conclusions:** Modifications to increase cultural relevance have the potential to improve fit and acceptance of a scale by the Indigenous population, however, these changes may decrease specificity and negative predictive value. There is an urgent need for suitable tools that are useful and reliable for identifying Indigenous individuals for clinical treatment of depression. This awaits future work for optimal specificity and validated cut-off points that take into account the high prevalence of depression in these populations.

## Impact statement

The present study suggests that modifying depression scales to fit the Indigenous context through changes to language or question structure is a culturally sensitive strategy that increases acceptance of psychological evaluation and treatment in communities. However, increasing acceptability must be balanced with maintaining clinical utility of instruments. The high prevalence of depression in these populations must be taken into account when developing culturally sensitive but specific tools.



## Introduction

Cultural safety is a combination of cultural awareness (acknowledging the differences between cultures), cultural sensitivity (respecting other cultures), and cultural competency (effectively working with diverse populations through appropriate behaviors, attitudes, measures, and policies; Marsella et al., 1985; Simon and Catherine, 2009). Research has shown that culturally competent care can improve communication between minority groups and health care professionals, including recognition of mental illness and assessment of its severity among minority groups (Schouten and Meeuwesen, 2006). These considerations are especially important in mental health care for Indigenous peoples, as Indigenous peoples share cultural backgrounds and ideas about health and concern that are unique from other ethnic groups (Mayberry et al.,

2000). As a response to these unique perspectives, cultural adaptation of psychometric tools has become a commonly used method to increase cultural safety of psychometric evaluation and to reduce communication problems (Gomez Cardona et al., Submitted). As a consequence, it is hoped to lead to reduced risk of harm, and to improved access of Indigenous peoples to psychiatric treatment for depression. Screening for symptoms with appropriate psychometric scales in community services is an approach to improve the quality of treatment for some Indigenous populations (Esler et al., 2007).

Measurement-based psychiatric care uses standardized measures to guide treatment and subsequently evaluate treatment outcomes (Aboraya et al., 2018). Validated psychometric measures can be used in first point of contact settings (i.e., primary care) to screen for psychiatric disorders, assess the need for treatment, monitor symptom severity, and track treatment outcomes (Porcerelli and Jones, 2017). Indigenous communities often lack specialized resources, including staff with specialized psychiatric skills (Boksa et al., 2015). Many studies have reported that there is a dearth of trained mental health workers of local Indigenous origin; moreover, a high turnover of non-Indigenous health workers leads to a lack of continuity of services and a lack of connection to specialized services or excessively long wait lists for Indigenous with severe mental illness (Boksa et al., 2015). This means limited access to an interview-based psychiatric diagnosis and specialist follow-up for relapses after treatment (Boksa et al., 2015). Moreover, trauma-informed care is rare, yet would be needed for safe and valid psychiatric assessment and interventions for Indigenous peoples. This is important as it is known that Indigenous people's health promotion and health seeking behaviors are largely influenced by a colonialistic past which has caused intergenerational social inequities; a lack of trust and confidence in many governments still prevails among Indigenous populations, making them less likely to be screened and causing an overall resistance to engaging with the healthcare system (Leung, 2016). Because compromised quality of mental health care is evident in many Indigenous communities, there is a need for effective symptom screening and monitoring with stable psychometric measures (Chan et al., 2021).

In addition to evaluating the need for treatment at the individual level, having culturally safe and accurate measurement of a population's mental health is essential for appropriate resource allocation and service planning at population and community levels (Chan et al., 2021). The use of psychometric screens at a community level can raise awareness of mental health needs and crises among stakeholders in Indigenous health and provide tools for evaluating the efficacy of interventions (Chapla et al., 2019). To make reliable conclusions on the efficacy of screens at community and population levels, it is important to ensure that psychometric tools are culturally safe and trauma informed, but also clinically useful, reliable when used with a specific population, and have a high quality as compared to traditional gold standards (Chan et al., 2021).

Standard qualities are requested from psychometric screens and outcome measures, and are also essential for culturally adapted depression scales. First, *reliability* represents a test's consistency across test questions (Andrade, 2018). Second, *validity* represents a test's accuracy such that test items are reported to be meaningful and relevant to the population they are used with (Andrade, 2018). Finally, the *clinical utility* of a scale indicates its utility for clinicians to diagnose and determine content of treatment (Labrique and Pan, 2010). In particular, the *sensitivity* of a screen indicates its capacity to correctly identify people who are most likely to benefit from a clinical diagnostic interview (Parikh et al., 2008). Accordingly, an optimal culturally adapted psychometric scale should not exclude from clinical interviews those who are depressed (*sensitivity*), but should also guide an efficient use of clinician resources for an interview with individuals who require treatment (*specificity*).

The prevalence of depression and anxiety, and incidence of suicide among Indigenous peoples, is commonly high in comparison to Western cohorts (Shen et al., 2018) It is known that a heightened presence of environmental stress and distress disproportionately raises the sensitivity of a diagnostic tool even where psychiatric care is not indicated or appropriate (Simon, 2015). As such, in Indigenous communities, the proportion of positive cases screened by a highly sensitive scale may misleadingly indicate a need to provide a clinical interview for the entire population (Parikh et al., 2008). Therefore, the optimal balance of different psychometric properties must be thoroughly tested in Indigenous communities before they are used to guide treatment.

Recently, more studies have explored different methods of culturally adapting and developing measures that reflect mental health conditions (Haswell et al., 2010). We recently reviewed the methods of cultural adaptation of measures for depression, identifying the modifications and adaptations made and evaluating their acceptability by target Indigenous populations (Gomez Cardona et al., Submitted). Here, we continue the work and investigate the psychometric properties of the previously identified culturally adapted psychometric measures. In this review, we assess the reported quality of the psychometric characteristics of adapted depression scales and their utility for psychological evaluation among Indigenous groups.

## Methods

The methods for the systematic search, including the search strategy, were reported previously (Gomez Cardona et al., Submitted). This study followed the outline set forth by the Preferred Reporting Items for Systematic Reviews and Meta-analyses (PRISMA; Figure 1). We searched the Ovid Medline, PubMed, PsycInfo, Embase, CINAHL, and Global Health databases for articles using three levels of search terms related to: a) Depression, b) Psychometrics, and c) Indigenous (Supplementary Material 1). After an initial search to capture articles from the inception of the databases to April 2021, we extended the search to the end of August 2022. Any reports with information on psychometric properties reported after extraction of these original studies were added to the current analysis as gray literature.

We extracted the information on utility of depression scales following cultural adaptation. This information included the "gold-standards" they were measured against, and the optimal cut-off point(s) determined after cultural adaptation (Table 1). We also extracted information on psychometric properties of adapted scales (Table 2), which were assessed through a quality criteria checklist (Supplementary Material 2). We extracted data on reliability (internal consistency, test–retest reliability, and inter-rater reliability), criterion validity (concurrent and predictive validity), construct validity (convergent and discriminant validity), cross-cultural validity (measurement invariance), incremental validity, and clinical utility (sensitivity, specificity, area under the ROC curve [AUC], positive predictive value [PPV], negative predictive value [NPV], and likelihood ratio [LR]). Detailed results on trends found across unique adaptation processes are presented in Supplementary Material 3.

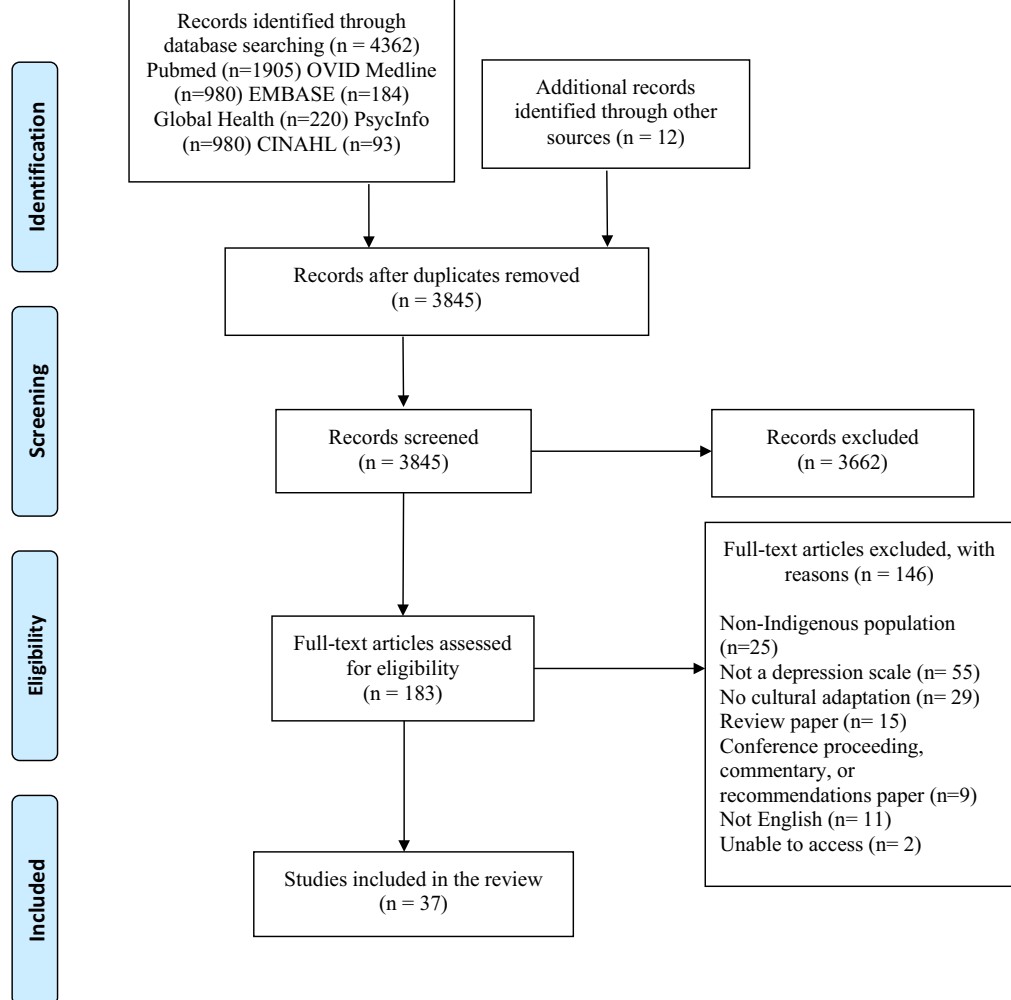

**Figure 1.** PRISMA flow diagram.

### Assessment of quality

We followed the guidelines of the ROBIS tool to assess the quality and risk of bias of this review (Whiting et al., 2016; Supplementary Material 4). Here, we report potential biases across several domains: a) study eligibility criteria, b) identification and selection of studies, c) data collection and study appraisal, and d) synthesis and findings.

### Results

#### Description of the adapted scales

Originally, we had identified 37 studies that met criteria in the systematic search (Gomez Cardona et al., 2021). Thirty-one (83.8%) of these studies reported results on the validation of their scales' psychometric propert(ies). Cohort sizes ranged from $n = 97$ to $n = 4,767$. Target Indigenous peoples were from native to Canada or the United States (5/31), Latin America (3/31), Asia (8/31), Africa (8/31), and Australia or New Zealand (7/31); many populations lived in rural settings. Each of the 31 studies produced a unique culturally adapted scale based on their individual methods. These 31 scales were, or are variations of, the following scales, listed in decreasing number of adaptations per scale: Center for Epidemiologic Studies Depression Scale (CES-D), $n = 8$; Patient Health Questionnaire (PHQ-9), $n = 7$; Edinburgh Postnatal Depression Scale (EPDS), $n = 4$; Kessler Psychological Distress Scale, $n = 4$; Hopkins Symptom Checklist Depression Scale (HSCL), $n = 3$; Geriatric Depression Scale (GDS), $n = 2$; Kimberley Indigenous Cognitive Assessment of Depression (KICA-dep), $n = 1$; Ndetei–Othieno–Kathuku scale (NOK), $n = 1$; International Depression Symptom Scale-General (IDSS-G), $n = 1$; Dar-es-Salaam Symptom Questionnaire (DSQ), $n = 1$; Kimberley Mum's Mood Scale (KMMS), $n = 1$; Self-Reporting Questionnaire (SRQ), $n = 1$; Functioning Assessment Instrument (FAI), $n = 1$.

#### Reliability

Cronbach's alpha was the most commonly analyzed psychometric property, reported by 77% (24/31) of adapted depression measurement scales (Ganguli et al., 1999; Tiburcio Sainz and Natera Rey, 2007; Bass et al., 2008; Campbell et al., 2008; Esler et al., 2008; Kaaya et al., 2008; Ekeroma et al., 2012; Gelaye et al., 2013; Armenta et al., 2014; Haroz et al., 2014; McNamara et al., 2014; Schneider et al., 2015; Bougie et al., 2016; Baron et al., 2017; Denckla et al., 2017; Haroz et al., 2017; Marley et al., 2017; Schantz et al., 2017; Gallis et al., 2018; Harry and Crea, 2018; Kilburn et al., 2018; Ashaba et al., 2019; Chapla et al., 2019; Hackett et al., 2019).

**Table 1.** Characteristics of culturally adapted scales

| Reference | Original scale | Adapted scale | Measured against ("Gold-standard") | Clinical use | Optimal culturally safe cut-off | Population that optimal cut-off is determined for |
|---|---|---|---|---|---|---|
| Armenta et al., 2014 | N/A | CES-D-20 | Diagnosis of MDD through the DSM-IV | ○ Negative affect and somatic difficulties items are useful to predicting MDD | N/A | ○ Young North American Indigenous (early to late adolescence) |
| Baron et al., 2017 | CES-D-20 | CES-D-10 | Diagnosis of major depression using MINI 6.0 (major depressive episode) | ○ Can indicate high risk of depression<br>○ Optimized cut-off is more acceptable for use in South Africa (compared to the Western cut-off)<br>○ Length is more feasible to use in research and clinical settings | 11–13 (mixed-results) | ○ Zulu, colored Afrikaans and Xhosa-speaking populations |
| Chapla et al., 2019 | CES-DC | Gujarati version of CES-DC | N/A | ○ Can be used to check prevalence of depression among youth | 15 | ○ School-going adolescents from Gujarati medium schools |
| Chapleski et al., 1997 | N/A | CES-D | N/A | ○ Is robust and useful to measuring depression factors (depressive affect, somatic, interpersonal, well-being) | N/A | ○ American Indian Elders over 55 years (urban, rural off-reservation, reservation) |
| Harry and Crea, 2018 | CES-D | Modified long and short form CES-D | N/A | ○ Has measurement invariance properties and can be used to assess mental health needs of marginalized groups | N/A | ○ American Indian young adults |
| Kilburn et al., 2018 | CES-D-20 | CES-D-10 | N/A | ○ Is a low resource screener for adolescent depression<br>○ Scores are related to individual and household determinants of depression (i.e., increasing age) | N/A | ○ Sub-Saharan African youth<br>○ Low socioeconomic status, rural communities |
| Tiburcio Sainz and Natera Rey, 2007 | N/A | CES-D-19 | N/A | ○ Are reliable measures of coping, emotional regulation, and negative affect | N/A | ○ Indigenous populations in Mexico |
| Schantz et al., 2017 | CES-D-20 | Short form CES-D | N/A | ○ Can measure nonspecific psychological distress and symptoms of clinical depression | 10 | ○ Indigenous in Andean Latin America |
| | PHQ-9 | PHQ-8 | | | 10 | ○ Patient population from urban and low-income setting |
| Caneo et al., 2020 | PHQ-9 | PHQ-2 | | ○ Comparable to the PHQ-9 in screening for major depressive syndrome | 3 | ○ Agricultural/rural Chilean population |
| Esler et al., 2008 | PHQ-9 | Modified PHQ-9 | Diagnoses of major depression if meeting all DSM-IV criteria; diagnosis of minor depression if meeting 2–4 of the DSM-IV criteria | ○ Is valid and acceptable as a part of clinical assessment for depression<br>○ Has stronger concordance with DSM-IV diagnosis criteria than other tools in this region | 9 | ○ Australian Aboriginals with ischemic heart disease |
| Gallis et al., 2018 | PHQ-9 | Urdu version of the PHQ-9 | Diagnosis of depression on SCID | ○ Is a valid tool to be used in low-income settings where depressive disorders are prevalent | 10 | ○ Pregnant women in Pakistan<br>○ Rural community settings |
| Gelaye et al., 2013 | PHQ-9 | Amharic version of the PHQ-9 | Diagnosis of depression on the SCAN | ○ Has high discrimination power and is comparable to diagnoses via clinical interviews | 10 | ○ Ethiopian and sub-Saharan African adults |
| Hackett et al., 2019 | PHQ-9 | aPHQ-9 | Diagnosis of major depressive episodes on the MINI 6.0.0 | ○ Has utility in assessment of depression levels, epidemiology studies, treatment monitoring, and outcome assessment | 10 | ○ Australian Aboriginal adults at high risk of depression<br>○ Rural and urban setting Primary care setting |

(*Continued*)

| Reference | Original scale | Adapted scale | Measured against ("Gold-standard") | Clinical use | Optimal culturally safe cut-off | Population that optimal cut-off is determined for |
|---|---|---|---|---|---|---|
| Husain et al., 2006 | PHQ | Urdu version of PHQ | Diagnosis of depressive disorder on the PAS | ○ Is reliable for screening depressive disorders ○ Can detect somatic and psychological problems | 5 | ○ Rural Pakistani population ○ People with little or no education |
| | SRQ | Urdu version of SRQ | | | 7 | |
| Bougie et al., 2016 | N/A | K-10 | N/A | ○ Can measure non-specific psychological distress (DSM-IV criteria for anxiety and mood disorders) ○ Can be used as a single factor score in the APS | 10–11 (mixed results) | ○ First Nations living off reserve, Métis, and Inuit |
| McNamara et al., 2014 | K-10 | K-5 | N/A | ○ Is a good measure for detecting likelihood of psychological distress ○ Can be used in population health surveys and clinical practice | N/A | ○ Older Aboriginal and Torres Strait Islanders Clinical setting |
| Mitchell and Beals, 2011 | K-10 | K-6 | Diagnosis of depression on CIDI | ○ Is an effective screener for psychological disorder ○ Can be used as the first stage in a two-stage sampling process that screens for potential cases before a lengthy diagnostic interview | 13 | ○ American Indians living on or near reservations |
| Fernandes et al., 2011 | N/A | K10 | Diagnosis of depression through the MINIPlus | ○ Are sensitive and specific instruments for screening during prenatal check-ups | 6 | ○ Prenatal women in South India Rural primary care setting |
| | | EPDS | | | 13 | |
| Campbell et al., 2008 | EPDS | Translated EPDS | N/A | ○ Can measure the same construct as the standard EPDS ○ Can identify participants at risk of depression better than the standard EPDS | 12 | ○ Aboriginal and Torres Strait Islander women (antenatal and postnatal) |
| Ekeroma et al., 2012 | EPDS | Samoan and Tongan languages translated EPDS | Diagnosis of serious depression on WHO-CIDI v3 | ○ Has strong internal consistency by language and ethnicity ○ Has greater reliability than the English version in this population ○ Can be used to detect postnatal depression prevalence in primary and secondary care settings | 10–17 (mixed results) | ○ Postnatal Samoan and Tongan women |
| Bass et al., 2008 | EPDS and HSCL-D hybrid | Scale to detect Maladi ya Souci | Diagnosis of a depression case by a local non-professional | ○ Can detect a local post-partum syndrome consistent with the DSM-IV diagnosis of MDD | 16 | ○ Women in the post-partum period in Kinshasa |
| Ashaba et al., 2019 | BDI-II; HSCL | "20-item depression scale" | Diagnosis of MDD through the DSM-IV, MINI-KID | ○ Can measure affective and cognitive symptoms of MDD | 10 | ○ Adolescents living with HIV in rural Uganda |
| Haroz et al., 2014 | HSCL; HTQ, AUDIT | MHAP-I | N/A | ○ Is reliable in assessing psychosocial problems and functioning | N/A | ○ Adult Burmese survivors of systemic violence in Thailand ○ Displaced community setting |
| Denckla et al., 2017 | NOK | Kiswahili and Kikamba translated NOK | N/A | ○ Can detect for depression, anxiety and somatic factors of depression | N/A | ○ Rural school aged children |
| Ganguli et al., 1999 | GDS | GDS-H | N/A | ○ Can detect cognitive and functional impairment symptoms of depression ○ Has higher distribution of scores in this population than what is reported from the original scale | 11 | ○ Illiterate elderly population in India |

**Table 1.** (*Continued*)

| Reference | Original scale | Adapted scale | Measured against ("Gold-standard") | Clinical use | Optimal culturally safe cut-off | Population that optimal cut-off is determined for |
|---|---|---|---|---|---|---|
| Sarkar et al., 2015 | GDS-15 | Tamil version of GDS-15 | Diagnosis of depression on ICD-10 | ○ Is reliable to give a point prevalence rate of geriatric depression<br>○ Indicates those at high-risk for clinical depression | 7 | ○ Elderly people in South India Rural community |
| Haroz et al., 2017 | HSCL-25 | IDSS-G | Diagnosis of depression on SCID | ○ Can detect functional impairment and suicidal ideation<br>○ Has utility for detecting DSM defined disorders<br>○ Can be used by non-specialist providers | 0.56 | ○ Non-Western populations Medical settings |
| Kaaya et al., 2008 | HSCL-25 | DSQ | N/A | ○ Is reliable to measure depression and anxiety<br>○ Can measure symptom constellation related to depressive disorder | 1.75 | ○ Indigenous Tanzanian population<br>Primary care setting |
| Marley et al., 2017 | EPDS | KMMS | Diagnosis of depression on DSM-IV | ○ Is a safe and reliable screener for identifying people at clinically high risk of anxiety and depressive disorders<br>○ Is more feasible for use than the original scale | 9 | ○ Aboriginal Australian perinatal women |
| Schneider et al., 2015 | N/A | FAI | Diagnosis of depression on MINI 6.0 | ○ Can be used to reliably screen maternal depression<br>○ Can complement or potentially replace (as a short instrument) other measures of functioning | N/A | ○ Woman who are pregnant or have young babies in Cape Town |
| Almeida et al., 2014 | N/A | KICA-dep | Diagnosis of a major depressive episode through ICD-10 and DSM-IV | ○ Useful to screen for presence of depression<br>○ High NPV is excellent at screening out non-depression cases | 8 | ○ Older Indigenous Australians |

Abbreviations: AUDIT, Alcohol Use and Disorders Identification Test; BDI, Beck Depression Inventory; CES-D, Center for Epidemiologic Studies Depression Scale; CIDI, Composite International Diagnostic Interview; CQ, Coping Questionnaire; DSM, Diagnostic and Statistical Manual of Mental Disorders; DSQ, Dar-es-Salaam Symptom Questionnaire; EPDS, Edinburgh Postnatal Depression Scale; FAI, Functioning Assessment Instrument; GDS, Geriatric Depression Scale; HIV, human immunodeficiency virus infection; HSCL, Hopkins Symptom Checklist Depression Scale; HTQ, Harvard Trauma Questionnaire; ICD-10, International Classification of Diseases-10; IDSS-G, International Depression Symptom Scale-General; K-5/6/10, Kessler Psychological Distress Scale 5-/6-/10 item; KICA-dep, Kimberley Indigenous Cognitive Assessment of Depression; KMMS, Kimberley Mum's Mood Scale; MDD, major depressive disorder; MHAP-I, Mental Health Assessment Project Instrument; MINI(-KID), Mini International Neuropsychiatric Interview (-Children and Adolescents); NOK, Ndetei–Othieno–Kathuku Scale; PAS, psychiatric assessment schedule; PHQ, Patient Health Questionnaire; SCAN, Semi-Structured Schedules for Clinical Assessment in Neuropsychiatry Interview; SCID, Structured Clinical Interview for DSM Disorders; SRQ, Self-Reporting Questionnaire; SRT, Symptom Rating Test; WHO-CIDI v3, World Health Organization Composite International Diagnostic Interview.
Scales: light blue: CES-D; dark blue: EPDS; light purple: PHQ; dark purple: NOK; light gray: FAI; dark gray: SRQ; light orange: KICA; dark orange: Kessler; light pink: HSCL; dark pink: KMMS; light turquoise: IDSS-G; dark turquoise: GDS; brown: DSQ.

**Table 2.** Psychometric properties of adapted scales

| Reference | Modifications | Reliability | | | Validity | | | | | | Clinical utility | | | | | |
|---|---|---|---|---|---|---|---|---|---|---|---|---|---|---|---|---|
| | | Internal consistency | Test–retest reliability | Inter-rater reliability | Concurrent (criterion) validity | Predictive (criterion) validity | Convergent (construct) validity | Discriminant (construct) validity | Cross-cultural validity | Incremental validity | Sensitivity | Specificity | Discrimination (AUC) | PPV | NPV | LR |
| Baron et al., 2017 | -h, t | α = 0.69–0.89 | | | | | r(FL) = 0.30–0.83 | | | | 71.4%–84.6% | 72.6%–95% | 0.81–0.94 | 16.1%–54.8% | | |
| Armenta et al., 2014 | +l, +s | α = 0.85–0.88 | | | | p < 0.001 | RMSEA = 0.04–0.08, SRMR = 0.04–0.08 | | | | | | | | | |
| Chapla et al., 2019 | t | α = 0.71 | | | | | r = 0.60, p < 0.001 | r = −0.25, p < 0.001 | | | | | | | | |
| Chapleski et al., 1997 | * | | | | | | r(FL) = 0.26–0.84, RMSR = 0.04–0.05, CFI = 0.86–0.93 | RMSR = 0.06, CFI = 0.89 | | | | | | | | |
| Harry and Crea, 2018 | +i | α = 0.84–0.87 | | | | | RMSEA = 0.03, CFI = 0.97, TLI = 0.97, WRMR = 0.99 | RMSEA = 0.02, WRMR = 2.10, TLI = 1.0 CFI = 1.0 | | | | | | | | |
| Kilburn et al., 2018 | -h, +s, t | α = 0.70–0.76 | | | | | r(FL) = 0.37–0.86 | RMSEA = 0.07, TLI = 0.89, CFI = 0.91, SRMR = 0.05 | | | | | | | | |
| T Tiburcio Sainz and Natera Rey, 2007 | f | α = 0.91 | | | | | r(factor) = 0.61–0.90 | | | | | | | | | |
| Schantz et al., 2017 | +i, +m, t | α = 0.81–0.83 | | | r = 0.76 | | r(FL) = 0.40–0.74, EVA = 3.21–3.44 | | | | | | | | | |
| Caneo et al., 2020 | t | | | | | | | | | | 74.6% | 93.9% | 0.92 | | | |
| Esler et al., 2008 | +a, f | α = 0.80 | | | | | | | | | 80%–100% | 12.5%–71.4% | | 30%–58.3% | 70.8%–95.2% | |
| Gallis et al., 2018 | +m, t | α = 0.84 | | | | | | | | | 94.7% | 88.9% | 0.96 | 75.2% | 97.9% | |
| Gelaye et al., 2013 | +s, +d, t | α = 0.92 | ICC = 0.92 | | p < 0.0001 | | r(FL) = 0.39–0.78, EVA > 1 | | | | 86% | 67% | 0.77 | 23% | 96.4% | 3.0 |
| Hackett et al., 2019 | +a | α = 0.88 | | | | | | | | | 81%–87% | 72%–82% | 0.88 | 43%–56% | 94%–97% | 2.8–4.6 |
| Husain et al., 2006 | t | | | | | | | | | | 69.6%–93.1% | 80.8%–85.2% | | 77.9%–78.9% | 77.8%–94.1% | |
| Bougie et al., 2016 | t | α = 0.84–0.88, r(item-item) = 0.21–0.71 | | | | | RMSEA = 0.05, CFI = 0.99, WRMR = 1.20–1.90 | | | | | | | | | |
| McNamara et al., 2014 | -h, +s, +m | α = 0.88–0.93, r(item-total) = 0.57–0.83 | | | | p < 0.0001 | | CFI = 0.99, TLI = 0.99, RMSEA = 0.04 | | | | | | | | |
| Mitchell and Beals, 2011 | * | | | | | | | | | p < 0.05 | | | 0.73–0.83 | | | |
| Fernandes et al., 2011 | t | | | | p < 0.01 | | | | | | 100% | 81.3%–84.9% | 0.95 | | | |

(Continued)

**Table 2.** (*Continued*)

| Reference | Modifications | Reliability | | | Validity | | | | | | Clinical utility | | | | | |
|---|---|---|---|---|---|---|---|---|---|---|---|---|---|---|---|---|
| | | Internal consistency | Test–retest reliability | Inter-rater reliability | Concurrent (criterion) validity | Predictive (criterion) validity | Convergent (construct) validity | Discriminant (construct) validity | Cross-cultural validity | Incremental validity | Sensitivity | Specificity | Discrimination (AUC) | PPV | NPV | LR |
| Campbell et al., 2008 | +m | α = 0.89–0.92, r (item-total) = 0.48–0.88 | | | p < 0.05 | | | | | | | | | | | |
| Ekeroma et al., 2012 | t | α = 0.86 | | | κ = 0.57 | | | | | | 80% | 80% | 0.83–0.89 | 82% | 86% | |
| Bass et al., 2008 | t | α = 0.76–0.88 | | | | | r = 0.34, p = 0.0001 | p < 0.0001 | | | | | 0.83–0.87 | | | |
| Ashaba et al., 2019 | +i, +l, +s, +a, t | α = 0.85–0.91 | | | | | r(FL) = 0.40–0.68 | | | | 81% | 78% | 0.84 | | | 4.2 |
| Haroz et al., 2014 | +s | α = 0.79 | r = 0.86 | r = 0.86 | p = 0.30–0.93 | | | | | | | | | | | |
| Denckla et al., 2017 | t | α = 0.77–0.84, r (item-item) = 0.17–0.53 | | | r = 0.15–0.45 | | r(FL) = 0.27–0.8, RMSEA = 0.04, CFI = 0.93; TLI = 0.92 | | | | | | | | | |
| Ganguli et al., 1999 | t | α = 0.92 | | | p < 0.05 | | r(FL) = 0.10–0.78, EVA > 1 | | | | | | | | | |
| Sarkar et al., 2015 | t | | | | | | | | | | 80% | 47.6% | 0.66 | | | |
| Haroz et al., 2017 | +l, +s, +a | α = 0.92 | r = 0.87 | ICC = 0.90 | p < 0.05 | | r = 0.56–0.78 | | | VIF = 2.8, p = 0.001 | 73%–77% | 67% | 0.72–0.75 | | | |
| Kaaya et al., 2008 | +d | α = 0.84, ICC = 0.89, r(item-total) = 0.29–0.76 | ICC = 0.82 | ICC = 0.89 | r(item-to-subscale) = 0.45–0.79 | | r(FL) = 0.24, EVA ≥ 1, r = −0.57 − −0.37 | r = −0.35 − −0.49 | | | | | | | | |
| Marley et al., 2017 | p, f | α = 0.89 | | | | | | | | | 82.6%–87% | 75%–86.8% | 0.86–0.90 | 54.1%–67.9% | 93.7%–94.4% | |
| Schneider et al., 2015 | * | α = 0.77 | | | | | p < 0.0 | | | | | | | | | |
| Almeida et al., 2014 | +i | ICC = 0.88 | | | | | | | | | 72–78% | 82–90% | 0.88 | 39%–50% | 96% | |

Abbreviations: AUC, area under the receiving operating curve; EVA, eigenvalue(s); ICC, intraclass correlation coefficient; FL, factor loading(s); LR, likelihood ratio; NPV, negative predictive value; PPV, positive predictive value.

Quality of values (Supplementary Material 2): red: poor psychometric qualities; yellow: moderate psychometric qualities; green: strong psychometric qualities.

Scales: light blue: CES-D; dark blue: EPDS; light purple: PHQ; dark purple: NOK; light gray: FAI; dark gray: SRQ; light orange: KICA; dark orange: Kessler; light pink: HSCL; dark pink: KMMS; light turquoise: IDSS-G; dark turquoise: GDS; brown: DSQ.

Modifications to scale: +/−, added/deleted; i, suicidal ideation items; h, hope items; l, loneliness items; s, somatic difficulty items; a, anger items; m, simplified language; d, local idioms of distress; p, changed scale administration protocol; t, translated; f, rephrased; *, other.

High alpha values were reported by all but four of the studies; moderate alphas corresponded with the CES-D, HSCL, and FAI (Haroz et al., 2014; Schneider et al., 2015; Kilburn et al., 2018; Chapla et al., 2019). These findings indicate good consistency in the types of questions used in the adapted measure as a whole.

Interclass correlation (ICC), item-item correlation, and item-total correlation are an alternative to alpha as a measure for reliability following adaptation. Six studies showed moderate properties of these metrics within the KICA-dep, K-10, EPDSb, NOK, DSQ, and K-5 (Campbell et al., 2008; Kaaya et al., 2008; Almeida et al., 2014; McNamara et al., 2014; Bougie et al., 2016; Denckla et al., 2017). Four studies demonstrated high properties of inter-rater and test–retest reliability (2 days to 1 week; Kaaya et al., 2008; Gelaye et al., 2013; Haroz et al., 2014, 2017).

### Cross-cultural validity

Measurement invariance testing was conducted in 4/31 studies (12.9%; Chapleski et al., 1997; McNamara et al., 2014; Harry and Crea, 2018; Kilburn et al., 2018). These tests used multigroup confirmatory factor analysis (CFA) to provide evidence for cross-cultural validity. These studies showed that some adapted scales were invariant across different groups of peoples, such as the same Indigenous Nation living in distinct residential locations (i.e., urban, rural off-reservation, and reservation; Chapleski et al., 1997).

### Criterion (concurrent and predictive) validity

Concurrent and predictive (criterion) validity was determined in 11/31 (35.5%) studies. Evidence for concurrent and predictive validity were reported through correlation analysis between scale ratings and ratings from the Diagnostic Interview Schedule or a well-established depression measure in the respective location (Ganguli et al., 1999; Campbell et al., 2008; Kaaya et al., 2008; Fernandes et al., 2011; Ekeroma et al., 2012; Gelaye et al., 2013; Armenta et al., 2014; Haroz et al., 2014; Denckla et al., 2017; Haroz et al., 2017; Schantz et al., 2017). About 7/11 studies (63.4%) measuring criterion validity demonstrated that the scale adaptations resulted in high levels of concordance with a psychiatric diagnosis of major depressive disorder (MDD) or a gold-standard measure of distress (Campbell et al., 2008; Kaaya et al., 2008; Fernandes et al., 2011; Gelaye et al., 2013; Armenta et al., 2014; Haroz et al., 2017; Schantz et al., 2017). However, 4/11 (36.4%) reports showed poor evidence for criterion validity (Kaaya et al., 2008; Ekeroma et al., 2012; Haroz et al., 2014; Denckla et al., 2017).

### Construct (convergent and discriminant) validity

Convergent and discriminant (construct) validity was measured through CFA, exploratory factor analysis (EFA), correlation analysis, or multivariate regression in 18/31 (58%) studies (Chapleski et al., 1997; Ganguli et al., 1999; Tiburcio Sainz and Natera Rey, 2007; Bass et al., 2008; Kaaya et al., 2008; Gelaye et al., 2013; Armenta et al., 2014; McNamara et al., 2014; Schneider et al., 2015; Bougie et al., 2016; Baron et al., 2017; Denckla et al., 2017; Haroz et al., 2017; Schantz et al., 2017; Harry and Crea, 2018; Kilburn et al., 2018; Ashaba et al., 2019; Chapla et al., 2019). The majority of these studies showed evidence for a high level of construct validity. Most of the adaptation processes improved the scales' ability to capture globally meaningful constructs of depression.

### Incremental validity

Only 2/31 (6.5%) studies examined incremental validity through regression modeling (Mitchell and Beals, 2011; Haroz et al., 2017). Strong incremental validity suggested that these adapted scales were able to predict the severity of outcomes among the Indigenous population better than existing measures. In the two studies, the predicted outcomes included lifetime mood disorders, physical diagnosis, alcohol use, and impaired functioning (Mitchell and Beals, 2011; Haroz et al., 2017).

### Clinical utility

Sensitivity and specificity were measured by 14/31 (45.2%) studies (Husain et al., 2006; Esler et al., 2008; Fernandes et al., 2011; Ekeroma et al., 2012; Gelaye et al., 2013; Almeida et al., 2014; Sarkar et al., 2015; Baron et al., 2017; Haroz et al., 2017; Marley et al., 2017; Gallis et al., 2018; Ashaba et al., 2019; Hackett et al., 2019; Caneo et al., 2020). Sensitivity and specificity are dependent on cut-off point(s) of the adapted scale. As such, 7/14 (50%) studies showed that the determined cut-off point of their adapted scales yielded high sensitivity (Esler et al., 2008; Fernandes et al., 2011; Gelaye et al., 2013; Baron et al., 2017; Marley et al., 2017; Gallis et al., 2018; Hackett et al., 2019), and in 7/14 (50%) cases, there was moderate sensitivity (Husain et al., 2006; Ekeroma et al., 2012; Almeida et al., 2014; Sarkar et al., 2015; Haroz et al., 2017; Ashaba et al., 2019; Caneo et al., 2020). In contrast, only 4/14 (28.5%) studies showed that the determined cut-off point yielded high specificity (Almeida et al., 2014; Baron et al., 2017; Gallis et al., 2018; Caneo et al., 2020), whereas in 10/14 (71.4%) cases, there was only low or moderate specificity (Husain et al., 2006; Esler et al., 2008; Fernandes et al., 2011; Ekeroma et al., 2012; Gelaye et al., 2013; Sarkar et al., 2015; Haroz et al., 2017; Marley et al., 2017; Ashaba et al., 2019; Hackett et al., 2019).

The cut-off score of an adapted scale also dictates the strength of the PPV, NPV, and AUC. These values must be strong in order to adapt scales to capture the true prevalence of depression among the Indigenous group. Fourteen (14/31, 45.1%) studies reported discrimination properties following cultural adaptation of the scale, determined by AUC (Tiburcio Sainz and Natera Rey, 2007; Bass et al., 2008; Fernandes et al., 2011; Ekeroma et al., 2012; Gelaye et al., 2013; Almeida et al., 2014; Sarkar et al., 2015; Baron et al., 2017; Haroz et al., 2017; Marley et al., 2017; Gallis et al., 2018; Ashaba et al., 2019; Hackett et al., 2019; Caneo et al., 2020). Of these, 4/14 (28.6%) scales had a strong discrimination and were able to discriminate between cases and non-cases of depression among the specific community (Fernandes et al., 2011; Baron et al., 2017; Gallis et al., 2018; Caneo et al., 2020). 9/31, 29.0%). Nine studies reported the PPV and NPV of the adapted instrument (Husain et al., 2006; Esler et al., 2008; Ekeroma et al., 2012; Gelaye et al., 2013; Almeida et al., 2014; Baron et al., 2017; Marley et al., 2017; Gallis et al., 2018; Hackett et al., 2019). Of these, only one scale had a high PPV (Ekeroma et al., 2012), but 8/9 (88.9%) had a high NPV (Husain et al., 2006; Esler et al., 2008; Ekeroma et al., 2012; Gelaye et al., 2013; Almeida et al., 2014; Marley et al., 2017; Gallis et al., 2018; Hackett et al., 2019).

Three studies reported the LR, and found moderate levels of LR following adaptation. (Gelaye et al., 2013; Ashaba et al., 2019; Hackett et al., 2019).

### Scale performance

Differences in adaptation methods across studies yielded differences in scale performance after adaptation (Table 2). There is evidence that the adaptation methods impacted the psychometric properties of the scales, evidenced by several patterns across scales. The adapted CES-D ($n$ = 8) was a particularly strong scale; there was evidence for its reliability, validity, and clinical utility across adaptations. The PHQ ($n$ = 7) performed poorly on specificity and PPV, however, it had excellent internal consistency, sensitivity, NPV, and construct validity. The Kessler scales ($n$ = 4) performed well across all validity tests and had high internal consistency. The EPDS ($n$ = 4) had good criterion validity, PPV, and NPV, but only moderate sensitivity, specificity, and discrimination. The HSCL ($n$ = 3) had good internal consistency and construct validity, but poor criterion validity and clinical utility metrics. The GDS ($n$ = 2) did not have high sensitivity or specificity. The NOK ($n$ = 1) had moderate internal consistency and high construct validity but poor criterion validity. The IDSS-G ($n$ = 1) had excellent reliability metrics and incremental validity, but did not have high specificity, and discrimination. The DSQ ($n$ = 1) had high reliability but poor validity. The KMMS ($n$ = 1) had high internal consistency, sensitivity, specificity, and NPV, but poor PPV. The FAI ($n$ = 1) had high construct validity and moderate internal consistency. The adapted KICA-DEP ($n$ = 1) showed acceptable internal consistency, specificity, and NPV.

### Discussion

In this review, we synthesized the global evidence for the psychometric properties of depression scales culturally adapted for Indigenous peoples. Many processes taken to develop and adapt such instruments were successful in improving the measures' reliability (internal consistency, test–retest reliability, and inter-rater reliability), convergent validity (construct and discriminant), and incremental validity. These processes included adding or deleting items, translation, and incorporating local idioms of distress. However, cultural adaptation methods had less success in improving criterion validity (concurrent and predictive) and cross-cultural validity (measurement invariance). Additionally, the adapted screening instruments were typically highly sensitive, which means they were useful for identifying individuals who might be depressed. Conversely, the specificity of many instruments was low. Despite most scales being acceptable among the population, for clinicians, it is possible that low specificity tools show no added value for screening within populations with a high prevalence of depression.

Some depression scales might be more globally suitable for use among Indigenous peoples than others. Among the studies, researchers chose 13 different original depression scales to adapt. Information on selection criteria and processes remain insufficient to conclude whether a higher number of reports indicates positive or negative characteristics of a scale. In this study, adapted CES-D outperformed other scales after adaptation, as it showed the most uniformly high psychometric properties.

### A gap in reporting characteristics

Testing for quality metrics was sporadic; only 2/31 studies tested for the majority of quality metrics (Gelaye et al., 2013; Haroz et al., 2017). Out of 15 psychometric characteristics to assess the effects of cultural adaptation, reliability and construct validity were two of the most commonly tested psychometric properties. In contrast, clinical utility metrics, including sensitivity, specificity, PPV, NPV, and LR, were less commonly measured. These results indicate a continued gap in the knowledge around the performance of adapted scales with Indigenous populations, namely, how different adaptation processes produce benefits for psychological testing in these communities. As such, we cannot draw strong conclusions about the clinical use of culturally adapted scales, nor about individual methods of cultural adaptation that maintain reliability and clinical utility.

### Increasing reliability and validity through cultural adaptations

Most reliability metrics of adapted scales were excellent, particularly in alpha values. This finding suggests that following adaptations, scale items were relatively consistent and that instructions for scoring scales were unambiguous. Yet, a limitation to using alpha to determine internal consistency is that higher alpha's do not necessarily indicate higher quality of a new scale (Sijtsma, 2009). Exceptionally high alpha values (>0.9) could have resulted when the adaptation process adds length or redundant items to the original scale, or when the modifications constrict the crucial constructs which are measured by the scale (Panayides and Walker, 2013). Some studies assumed higher reliability of the developed scale following a high alpha value. In some cases, researchers even relied on alpha values to remove items that did not correlate well; however, this is not the developed purpose of alpha values (Cartagena-Ramos et al., 2018).

Most studies did not cover multiple types of validity. Based on few positive results, some adaptation processes improved understanding and acceptance among the Indigenous population. Many scales had high construct validity, showing that factors of the scales represented true constructs of depression known to the specific Indigenous group, such as affective or somatic symptoms. A few studies also found acceptable criterion validity, showing that the scales accurately responded to established criterion of depression used by gold-standard instruments with populations where the original scale is used. Measuring incremental validity of scales was important to understand if adapting scales, such as through incorporating local idioms of distress, predicted outcomes above and beyond previously established Western measures. A few scales with high incremental validity were useful to predict functional impairment above the scores on non-adapted measures; they showed which scores most accurately indicated mental health concern in that population.

### Measurement invariance

A scale's measurement invariance, or cross-cultural validity, indicates how well a new scale minimizes the inter-rater differences, such as how different populations endorse scale items (Bader et al., 2021). Some scales proved to be measurement invariant following adaptation processes, meaning different cultural groups interpreted the constructs of depression in the scale in a conceptually similar way. Unfortunately, most studies did not compare the results of the scale's properties between Indigenous and non-Indigenous groups, or between different Indigenous groups (Bougie et al., 2016; Kilburn et al., 2018). In fact, without invariance testing, results might not even generalize to the same Indigenous group living on-reserves or in rural areas (Bougie et al., 2016). Therefore, it remains inconclusive whether these adaptation processes increased cultural safety without sacrificing the accuracy of the original scale. An adapted

scale should ultimately balance measurement invariance and diagnostic ability for it to be useful.

## Clinical utility

For clinical utility, the fact that the prevalence of depression is high in many Indigenous communities should be considered in future studies. Increasing the acceptance of a screening tool through cultural adaptation may inadvertently increase its sensitivity by lowering the threshold to be considered a positive case (Shen et al., 2018). Specifically, we found that the sensitivity of adapted scales was often considerably higher than its specificity. Accordingly, PPV and NPV represent the percentage of individuals who truly do or do not have a depressive disorder (respectively), and a trend of high NPV and low PPV among the scales may be explained by the prevalence rates (Simon, 2015). The findings suggest that processes to culturally adapt scales can make them more effective at detecting positive cases of depression rather than screening out negative cases in the population (Labrique and Pan, 2010).

Moreover, a pattern of low PPV suggests the added or modified scale items may not have been disorder specific after cultural adaptation. Our results on the NPV and PPV of adapted scales align with previous literature stating that predictive values are one of the most important metrics to guide treatment of clinical practitioners (Labrique and Pan, 2010). Predictive values indicate the diagnostic capability of the test in the real-world and are thus referred to as the scale's *real-world performance* or *clinical relevance* to screening individuals who may benefit from a more accurate diagnostic interview and treatment guided by a diagnosis (Labrique and Pan, 2010). Although predictive value metrics were seldom reported by studies, studies have stated that this psychometric evaluation should not be forgotten. Predictive value scores reflect congruity with Western-based ideas about the definition of depression, how it manifests and should optimally be treated; community knowledge may enhance clinical utility and safety in presentations that do not match Western ideas (Haswell et al., 2010).

The studies indicate that to increase the *clinical relevance* of the adapted scales, it is necessary to increase NPV (minimizing false negatives), and to increase PPV (minimizing false positives). The findings in the original studies suggest that to increase NPV after adaptation, it is best to avoid or exclude scale items which are likely to be endorsed easily by the entire population (i.e., not measuring a locally specific symptom of depression) as may innacurately increase the sensitivity of the scale for a true depressive case (Mitchell and Beals, 2011; Sarkar et al., 2015; Simon, 2015). Additionally, to increase PPV and to increase specificity, it is recommended to focus questions on motivational, cognitive and affective components of depression rather than on symptoms that may arise from non-psychiatric medical conditions, as these can inflate scores and make the scale less specific for a depressive case (Ganguli et al., 1999; Simon, 2015). The studies noted that increasing specificity is particularly necessary in low-resource settings, such as the settings where most adaptations were conducted. The value of a screen with high specificity in Indigenous communities is that it allows clinicians to effectively allocate resources such that the most in need receive timely treatment.

The AUC of a few studies was high, showing that the cut-off points determined after cultural adaptation had discriminative capacity (Schwarzbold et al., 2014). However, because few studies assessed AUC, we cannot reach a strong conclusion on the discrimination properties of most types of adapted depression scales. Similarly, the LRs of most adapted scales in the studies were

moderate, but there is a lack of data to reach conclusions on their ability to screen for probable depression. A high LR is the likelihood for having a high risk for depression – as indicated by the cut-off point of the adapted scale – for a person with current MDD compared to someone who does not (Hackett et al., 2019).

## When could a cultural adaptation be useful?

Ideally, cultural adaptations would be useful when balancing two goals: a) increase acceptability through translating Western constructs of psychological problems to the Indigenous context, and b) support treatment (Kohrt et al., 2014; Chan et al., 2021). The strength of cultural adaptation is that it can improve scale items' reliability and validity through changes to language or representation of depression constructs. At the same time, there are limitations inextricably linked to these changes, including their impact on clinical utility.

Symptom measures are a Western concept, and the Western medicine has been built to treat individuals based on symptom severity (Bredström, 2019). Doctors need certain criteria to evaluate the need for treatment and to define recovery. Researchers have also used scale cut-offs for intake and to define how many benefitted from a certain intervention. An existing gold standard for a diagnosis is essential for validation after cultural adaptation, and to validate diagnostic cut-offs for a screen. Without tools available to have an accurate prevalence measure or diagnostic tool, it is difficult to ascertain true clinical utility of adapted scales. Defining a gold-standard for depression is harder, and a gold standard measure is often not available, in Indigenous areas (Kaaya et al., 2008; Haroz et al., 2014).

## Future lines of work

Action is needed to address knowledge gaps around Indigenous mental health constructs as well as to understand how interventions, policies, or programs can support unmet needs of these populations. For this to be possible, there must first be an understanding about Indigenous concepts of emotional or mental distress and wellbeing. This forms a basis for treatments as well as tools to identify individuals and communities in need of support and interventions. However, in some instances, cultural adaptation might not be the primary approach in building trust and supporting empowerment (Gomez Cardona et al., 2021). As an alternative to cultural adaptation, methods of administration could also increase cultural safety. This includes at least considering the setting, language, use of community members at administration, and visual elements (Gomez Cardona et al., Submitted).

It is not rare for Indigenous communities to question the concept of a symptom focused approach that uses distinct cut-offs (Gomez Cardona et al., 2021). In fact, research teams have worked to co-design culture-specific tools for a culturally based, often community-targeted interventions supporting empowerment rather than symptom reduction to meet cut-offs (Haswell et al., 2010; Gomez Cardona et al., 2021). Thus, if you accept a non-symptom-based approach for interventions, you will not need symptom-based measures. This should be a main consideration when evaluating the necessity to culturally adapt a depression measure for a particular Indigenous group.

In the future, studies should include a needs assessment prior to developing new screening tools. If modifications are warranted, qualitative methods may be beneficial to understand the community's needs which can be addressed by a novel or adapted scale

(Gomez Cardona et al., Submitted). Concurrently, there must be emphasis on ensuring adaptation processes yield stable and clinically useful tools. This represents a shortcoming observed across the studies in this review; there was limited comprehensive testing of multiple psychometric domains. It is necessary to not only evaluate reliability and validity following changes to the scale, but also the utility of the new scale to accurately detecting the risk of depression.

Similar to what has been found by other researchers who adapted or developed new scales for use with Indigenous peoples, we advocate for future studies to examine scales' sensitivity to change (Haswell et al., 2010). Sensitivity to change reflects the competency of the scale to detect changes in mood, and as such, is an essential characteristic to evaluate treatment response. Evaluation and treatment of mental health relies on stable and measurement invariant tools for screening those in need of treatment, which should simultaneously signal the direction and magnitude of effect sizes of treatments to understand their efficacy for a particular population (Fitzpatrick et al., 2019).

## Limitations of the current study

Five databases were systematically searched. Hence, it is possible that not all relevant studies were identified through systematic search strategies. To reduce the risk, gray literature found through hand-searching was incorporated in the review; this includes searching on open access repositories and through checking references of included articles. Several key terms do not have globally accepted terms, most importantly, *Indigenous* or *cultural adaptation,* which may have limited the scope of our screening. The conclusions made for validity and utility are limited by the fact that the included reports on cultural adaptation described a limited number of characteristics. This limits this study's ability to draw strong conclusions to guide a specific selection of tools for clinical use or for research.

We only considered studies published in the English language. This ensured that data for the studies' validation analyses were uniform, however, it is possible that some existing adaptation processes were left out of this review if they were reported in a different language. Further validation, analysis, and adaptation of new and existing measures of depression is needed and will confirm the applicability of such instruments in culturally distinct populations.

## Risk of bias

The four domains of ROBIS were completed, which indicated this review was completed with a low risk of bias. This pertains to study selection, data collection and study appraisal, and data synthesis. The conclusions of the review are supported by the evidence presented and included consideration of the relevance of included studies. The methodology of the synthesis was driven by the nature of the studies and our research objectives, however, since a meta-analysis was not conducted, no statistical synthesis methods were undertaken.

## Conclusion

Through a review of the literature, we found evidence that cultural adaptation may increase the validity of depression scales and their reliability to be used in mental health assessment of Indigenous populations across the world. The current review supports the use

of adapting scales to fit the Indigenous context, increasing its acceptability to community members and overall consistency. At the same time, psychometric testing of adapted scales highlights a potential caveat of losing clinical utility with too high sensitivity and low specificity. Cultural adaptation of depression assessments for Indigenous populations would be clinically useful when balancing two goals: 1) increase acceptability through translating Western constructs of psychological problems to the Indigenous context, and 2) support treatment.

**Open peer review.** To view the open peer review materials for this article, please visit http://doi.org/10.1017/gmh.2023.52.

**Supplementary material.** The supplementary material for this article can be found at http://doi.org/10.1017/gmh.2023.52.

**Acknowledgments.** We would like to thank Ms. Andrea Quaiattini from the McGill University Library for her assistance with our comprehensive systematic search procedure. We also express our gratitude for Indigenous co-authors in other publications and some other community members in Montreal and Kahnawà:ke for introducing us to some basic concepts of their view of health and wellbeing as opposed to symptom measurement.

**Author contribution.** Conceptualization: L.G.C., O.L.; Formal analysis: M.Y., Q.S., L.G.C., V.N., M.K., G.V., O.L.; Funding acquisition: O.L., L.G.C.; Investigation: M.Y., Q.S., V.N., M.K., O.L.; Methodology: L.G.C., Q.S., M.Y., O.L., V.N.; Project administration: O.L., Resources: O.L.; Supervision: O.L.; Validation: O.L.; Visualization: M.Y., O.L.; Writing – original draft: M.Y., Q.S., L.G.C., O.L.; Writing – review and editing: M.Y., Q.S., L.G.C., V.N., M.K., G.V., O.L.

**Financial support.** O.L. was funded by grants from FSISSS (#8400958), CIHR Institute of Indigenous Peoples' Health (#426678), FRSQ (#252872 and #265693O), and the Réseau Québécois sur le suicide, les troubles de l'humeur et les troubles associés. L.G.C. had financial support from the Réseau universitaire intégré de santé et services sociaux (RUISSS) McGill and CIHR Institute of Indigenous Peoples' Health (#430331). Q.S. had funding from McGill University's Healthy Brains Healthy Lives fellowship. The funders had no role in study design or in later collection and data analysis.

**Competing interest.** The authors report no competing interests. The authors alone are responsible for the content and writing of the paper.

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
