## [Reviewer Report]

We submit our systematic review “Safe and valid? A systematic review of the psychometric properties of culturally adapted depression scales for use among Indigenous populations” for consideration in your journal. This complements our scoping review on methods of cultural adaptation of depression scales. The present study suggests that modifying depression scales to fit the Indigenous context through changes to language or question structure is a culturally-sensitive strategy that increases acceptance of psychological evaluation and treatment in communities. However, increasing acceptability must be balanced with maintaining clinical utility of instruments. Considering the high prevalence of depression in these populations must be taken into account when developing culturally sensitive but specific tools. 

Yours,

Outi Linnaranta, MD, PhD

Chief Physician 

Finnish Institute for Health and Welfare

Helsinki, Finland

---

## [Reviewer Report]

Thank you for the opportunity to review the manuscript ‘’Safe and Valid? A Systematic Review of the Psychometric Properties of Culturally Adapted Depression Scales for Use Among Indigenous Populations.’' This well-written manuscript seeks to expand the literature on the cultural safety and validity of adapted depression scales for Indigenous populations. Using a systematic review following the PRISMA guidelines, the authors synthesize and summarize the global evidence for the psychometric properties of various depression scales culturally adapted for Indigenous populations.

Reviewer’s Comments:

Strengths

The manuscript will significantly contribute to the field with a high impact.

The manuscript is original and fills a gap in the literature.

The manuscript covers global content, including research inclusion, presentation of results, and discussion.

The findings will contribute to advancing knowledge in the field.

The authors convey their ideas and present their results in an organized and structured manner.

Areas for Improvement

The authors could have provided more details in the manuscript’s methodology.

A PRISMA flow diagram would be helpful.

There are some areas where the manuscript could benefit from improved clarity or more concise language.

1. In the conclusion section of the abstract, the statement indicating a reduction in specificity and negative predictive value does not logically follow the results presented in the abstract.

2. On Page 5, line 9, the statement ‘’that are unique to other ethnic groups’' suggests shared health beliefs with other ethnic groups rather than a unique perspective.

3. Page 5, line 12 – would better psychometric evaluation improve healthcare access, as stated, or providers' ability to diagnose/treat?

4. Page 6, line 28 - The text ‘’for mental health crises amongst stakeholders’' is confusing as written and suggests that awareness is for mental health crises among stakeholders.

5. Page 6, line 31 – How should measures be comparable?

6. Page 9, line 103 – Identifying the instruments with low alphas would be helpful.

7. Page 12, line 178 – As written, the authors suggest that all adapted scales performed similarly. Clarification would be helpful.

Many references were published more than ten years ago – I recommend updating references where possible.

---

## [Reviewer Report]

This is an important piece of work, as for a very long time, there was little scrutiny of how depression scales were working among Indigenous populations. I make two comments on the strength and importance of the paper.

Distinguishing between an expected loss of social and emotional wellbeing due to ongoing grief and loss, adverse life events and heightened inequalities in social and environmental determinants of health vs clinical depression is critically important. Practitioners working cross-culturally need tools that can be relied on to decide whether to refer a client in distress to social and emotional wellbeing support only or to also refer to a mental health clinician for medical support and determine the level of urgency.

This study identified 13 tools in the literature that have been culturally adapted to support health workers and clinicians be able to do this, and revealed that their reliability, sensitivity, validity and predictive accuracy limits is infrequently determined. Its findings that some scale properties are enhanced by cultural adaption while others are decreased allows readers to understand potential strengths and limitations of adapted tools. The paper provides an important message to health services that the psychometric properties of the tools they select for the diagnosis of depression are important, and that other processes in the clinical pathway, such as stronger decision support tools may be needed to assist anyone experiencing distress but better identify those who would benefit the most by seeing a mental health clinician. This is well described in the paper’s discussion.

I suggest the authors may explore the following document in the grey literature that was not identified, probably because it was made available through an internal mental health clinicians’ part of a state health department for 10 years and downloaded hundreds of times, and more recently was placed on a site open to the public here (https://apo.org.au/node/19597). A modified Kessler (K6+2) was used here and its psychometric properties provided in Haswell et al., 2010; validation of the Growth and Empowerment Measure. However, I wish to strongly identify a COI here and leave it to the authors’ decision whether is meets the inclusion criteria as I am the lead author.

One very important quality not mentioned here is sensitivity to change. I am conscious of that because in my own work, Kessler scale is very insensitivity to change because it bounces around wildly between intervals and I think reflects the external environment and completely normal responses to the external environment, hence a poor reflection of the effectiveness of treatment. The need to assess this could be mentioned in the Discussion for further work.

Mostly minor suggestions are below. Many are just small things about writing in the introduction, that I think could convey the meaning and importance more clearly for readers.

Impact Statement page 1 = the last sentence is incomplete, suggest deleting the word Considering.

Abstract Background –

Suggest the aim be stronger, not just to summarise but to critically examine, interrogate perhaps? Also suggest “However, the published findings on psychometric properties …”

The last sentence of the conclusion would also be reworded – eg starting with “There is an urgent need to…”

Introduction

Paragraph 1 = I suggest the authors consider placing the second sentence – Culturally (safe and) competent care can increase communication … Cultural safety is a combination … (authors disgression)

Perhaps include the word trauma-informed as well, some of the tools used in clinical settings have little regard for their potential to re-traumatise or make people feel worse, this would also go with the title of ‘Safe and valid?’

…suggest include: and to reduce communication problems (that interfere with accurate assessment).

Last sentence – perhaps clearer to state:

As a consequence, cultural adaption aims to improve access (and reduce the risk of harm) ….

Para 2 = Indigenous communities often lack (culturally safe and appropriate resources) …

(sometimes is not specialisation that’s needed, problems can rise with too much specialisation)…. This (can) result in limited access to interview-based ….

Line 28 = at community level can raise awareness (of) mental health (needs and) crises…. (However,) to make reliable …

Line 31 = are culturally safe [maybe trauma informed?] and clinically useful, reliable (when) used …

Line 33 = (Many) --- [several sounds more like three] standard qualities are required of psychometric screens and outcome measures (which are also essential for)

Line 36 = which (are) meaningful and relevant

Line 37 = clinical utility of a scale [omit depression as you seem to be speaking generally about these qualities] (assesses) its use by clinicians to diagnose and determine treatment.

Line 39 = identify people who are most likely to benefit from a clinical diagnostic interview …

Line 42 – suggest a new paragraph – The prevalence [delete rate as not a function of time] of depression and anxiety and incidence of suicide … Heightened presence of illness and suicide risk disproportionately raises the sensitivity [and reduces the specificity?]

***on this point, I think it is important to be very clear. Remember that whole communities can be at risk, not just individuals, so you don’t want reduced sensitivity in clinical diagnosis unless you acknowledge that people with less severe clinical depression be left unidentified and untreated, but this leaves the possibility that the depression may then progress to more severe manifestations if not assisted well – so need to be very careful in meaning. Reduced specificity in diagnosing an clinical mental illness is more problematic – as the ability to distinguish a loss of social and emotional wellbeing which could be best supported by community if possible rather than diagnosis and medication. Need to be very careful here.

(again the document above https://apo.org.au/node/19597 suggests ways to address that in primary health care practice in Australia).

Line 50 = you may want to say “developing measures that reflect mental health conditions” not just depression.

Line 54 = we assess the (reported) quality …

Line 201 = see above, Line 42. Is this suggestion about clinicians coming from clinicians themselves? I would again be cautious suggesting that high sensitivity is not a desirable quality, it is the low specificity that is the problem.

Line 203 = I don’t understand the sentence starting, “Yet… please clarify.

***Line 233 and Section 4.4 = very good points made here, predictive value scores reflect congruity with Western-based ideas about what is depression and how does it manifest.

One could argue that anger (measured in K6+2 Haswell et al., 2010) has been ignored in depression by Western psychiatry – but is a prominent emotional response to continuous grief and loss and injustice that can mask depression (and possibly lead to suicide) – in this case, community knowledge may enhance clinical utility (safety) in presentations that don’t match Western ideas.

The rest of the paper is also excellent and clear.

---

## [Reviewer Report]

Please find enclosed our revised article “Safe and valid? A systematic review of the psychometric properties of culturally adapted depression scales for use among Indigenous populations”, which we resubmit for consideration in Cambridge Prisms: Global Mental Health. 

Yours,

Outi Linnaranta, adjunct professor

McGill University, Canada